# Course Crafting in the Pandemic: Examination of Students' Positive Experiences

Lakshmi Narayanan [1], Ramzi Nasser [2,*] , Shanker Menon [1] and Brett Wallace [3]

1   Department of Psychology, University of North Florida, Jacksonville, FL 32224, USA;
    l.narayanan@unf.edu (L.N.); s.menon@unf.edu (S.M.)
2   College of Education and Arts, Lusail University, Lusail P.O. Box 9717, Qatar
3   College of Liberal Arts and Social Sciences, Central Michigan University, Mt Pleasant, MI 48859, USA;
    lwsnwallace@gmail.com
*   Correspondence: rnasser@lu.edu.qa

**Abstract:** College students' positive experiences were examined before and during COVID-19 when courses transitioned to a new format. During COVID-19, university courses at a state university in the US transitioned to a new online format. This study observed the affective experiences of students through the instructors' course crafting during times of crisis. The method of critical incident qualitative data collection examined what students perceived as positive experiences. Students' perceptions were examined, and the nature and types of positive experiences were examined before and during the pandemic. This comparison provided insights into the emotions and feelings experienced between the two groups (Before COVID-19 and During COVID-19). As courses were modified and redesigned for remote learning, examination of the findings showed students' positive experiences considering the faculty recrafting their courses, which gave a valuable insight into the dynamics of these positive experiences in the teaching and learning process. Educators crafted their courses' changing format and provided emotional support, empathy, kindness, reassurance, and encouragement when needed. The implications of these findings and understanding how we can thrive and flourish even in very challenging times in the virtual environment is discussed.

**Keywords:** course crafting; positive psychology; COVID-19 (during and after); positive experiences; intensity of the experience





## 1. Introduction

A new reality has emerged after the COVID-19 epidemic began in 2020. The spectrum of education delivery and the way educators deal with students has changed drastically. While COVID-19 has had an impact on practically every facet of our life, it would be important for researchers to study not just the negative but the positive events in the educational environment during the pandemic. Schools and institutions had to switch overnight to a new mode of learning. The "new normal" consisted of conversations and interactions conducted while "plugged in". Consequently, endless hours of teaching and learning shifting to a virtual format [1].

Few warnings were given to staff and students of the online shift, and many of the digital pedagogies that faculties adopted for teaching and learning were done so to satisfy the immediate needs. Instantaneously, scholars were researching these changes in schools and colleges, looking at how they affect teaching and learning as well as overall academic work behaviors and pedagogical transformation. Given the conditions of uncertainty, many institutions were looking at highly flexible and adaptable options. This included changing various guidelines for courses where instructors, students, and administrators responded to the need of examining the data on teaching and learning switching to virtual education [2–4]. Such initiatives were initiated to understand key and affective interactions between educators and their students.

With the advent of COVID-19, academics adjusted to online interactions and teaching methods. They were forced to recognize and adopt to new technologies, new teaching methods, and sequencing the curriculum to meet emerging challenges of being "plugged in". Bates [5] made several important recommendations and suggestions on the "dos and don'ts" of instruction and learning in this pandemic environment. Bates highlighted dos that included educators needing to understand the technologies in order to engage online students to increase their use in the classroom. A significant don't was that instructors employed the most practical and simple instruments available at a critical period of a paradigmatic shift to avoid having to alter their teaching strategies. However, in the long run, instructional strategies must adapt to the unique needs of online students. Also, in the rush of online "dos", a psychological dimension appeared in the pandemic; this included increased isolation with the risk of loneliness, limiting people's access to normal social support networks, and weakening coping mechanisms. These isolation effects and the psychological impact of isolation can persist for months or even years before they improve [6,7]. It was suggested that it is possible to apply positive psychological interventions as a promising strategy that has been demonstrated to increase wellbeing across wide demographic differences and personality dispositions [8]. Positive psychological interventions, if put to practice, could cultivate positive feelings, such as "savoring", "kindness", "empathy", "optimism", and "satisfaction".

### 1.1. Research on Positive Psychology during Trying Times

According to the perspective of positive psychology, positive interventions can help in comprehending what it means to thrive in life [9]. The contention is that most of the existing conventional psychological research has mostly concentrated on the negative results and "what goes wrong", while positive psychology emphasizes the importance of comprehending "what is good" in life [10].

A plethora of studies have examined the detrimental effects and repercussions of COVID-19, including emotional discomfort, melancholy, anxiety, and anger [11,12]. Other research has examined elements including irritability, rage, avoidance tendencies, mental health problems, and other risk factors [7]. The positive aspects of the COVID-19 experience have lately been addressed in various studies [13,14]. Some scholarship has underlined human affinity in the context of the pandemic [15] and the maintenance of positive attitudes in the face of trying times, which made students be able to adapt and become resilient in the face of social isolation [16]. Several researchers have examined the stress and coping mechanisms of teachers and students and the use of positive psychology in their engagement in a learning community [13]. In a Japanese context, the function of positive emotions and their relationship to mental health during the COVID-19 pandemic was studied [14]. It is suggested that the maintenance of a positive attitude in the face of trying times and experiences could help students become more resilient They also contend that these situations open opportunities enabling educators to ponder existentially on the jobs or tasks at hand.

### 1.2. Positive Psychology, Course Crafting, and the Pandemic

In the field of positive organizational psychology, a method that has become popular is job crafting, a process used to cultivate positive meaning and identity in the experiences people face at their work. Job crafting is a process which involves essentially changing tasks, relationships, and perceptions [17]. We looked at course crafting as a process by which courses are modified or "crafted" to deliver effective outcomes for students based on the job crafting model. Job crafting was first introduced by Wrzesniewski and Dutton [18]; they described how employees can become "crafters" of their jobs by doing it three ways: task crafting, relationship crafting, and cognitive crafting. Task crafting is modifying or altering the tasks to make them more meaningful. For example, task crafting here would be changing some assignments to relate them to the COVID-19 experience, where students can reflect on their personal experience and be able to share it in a community of learners

and connect to current knowledge and common applications in the field. An example of relationship crafting could be changing how students interact with each other and the instructor. They could have more Zoom office hours where teachers interact directly with students, and they could do group assignments and work online together. Examples of cognitive crafting or thought crafting involve ways to change student perspective to be more meaningful; for example, there could be an assignment where students could empathize with others in relation to what life is like in the pandemic by comparing themselves to others less fortunate. Job crafting can create more meaning and motivate and engage participants. Job crafting is also about coping and being resourceful in adapting, adjusting, and being flexible on the job.

Some researchers have suggested that many employees may proactively seize the opportunities and resources needed to cope with specific situations to craft their jobs and take responsibility in order to fulfill the expectations and needs of their work [19]. In this paper, we look at this same model and try to understand how individuals used opportunities and resources to cope with specific situations, in this case, the pandemic, and take responsibility to fulfill what was required of them for this situation. We understand that job crafting takes place at the organizational level and although this is a very different situation to the one discussed here, we just try to use their framework in terms of the different types of ways people craft jobs in order to understand the crafting process for these courses and the student perceptions and reactions to these changes.

The COVID-19 pandemic presented a unique opportunity to use the job crafting model as academics transitioned to the virtual format to examine the positive experiences of students, by focusing on how courses were crafted. The course crafting process naturally occurred, as instructors were crafting before the pandemic and quickly had to recraft their courses. In the pandemic, content developers, course designers, technical experts, counselors, and advisors to students worked and put together courses to transition into a remote format. Crafting brought changes, and we wanted to understand how these changes worked in a time of crisis to make teaching and learning more meaningful, motivating, and engaging for students in a new virtual format [20,21].

In this study, we looked at how courses were crafted and what features of this redesign of courses enhanced, engaged, and made the course more meaningful, and promoted positive and affective environments for students. The effectiveness of course crafting was measured with student perceptions of positive experiences during the pandemic. We applied the concept of job crafting to "course crafting", since courses had been collaboratively crafted by different instructors to meet common standards and student learning outcomes in time of crisis. Extension and elaboration of the job crafting model has been also used by other researchers. For example, Leana et al. [22], expanded the job crafting model to examine relational crafting when used in childcare centers to meet shared objectives and understand the effectiveness of this process in the childcare classroom. Since the goal and objective of this crafting was to meet the needs of students in the learning process, we attempted to measure the outcome of satisfaction and engagement with the course by looking for students' positive experiences.

### 1.3. The Present Study

This study began exploring positive experiences of students for an on-going study before the pandemic of COVID-19; the time of the study coincided with the period of the pandemic, which gave us a unique opportunity to understand what changed for students as courses became remote. We examined what the course crafting elements that students responded positively were.

The positive psychology perspective was used as an overarching framework driving this study. In stressful and challenging situations, it is important to understand how individuals savor their positive experiences and use them to reduce the stress and anxiety created by the situation in an environment with extreme isolation [23]. The examination of

how and what students perceived as positive experiences was conducted using a critical incident method.

We used a critical incident method, which usually has been used to evaluate the performance of an employee's actions in specific situations in an organization. This method specifically examines incidents to identify the experiences within the online educational context, or instances during a period that made students feel positive. This method is detailed in the Materials and Methods section.

The critical incident method identified and perceived positive incidents among students. The context of the study, which was during a turbulent period and a very precarious time of uncertainty, gave us a glimpse of the students' emotional status.

The qualitative data helped us to understand the positive psychology-related conditions and the perceptions of students regarding the course crafting of their faculty. This study is unique in that it focused on positive experiences as courses were being delivered remotely. A comparison was made of the positive experiences of students before and after the course crafting over the pandemic period, as we had data to compare the positive experiences before the pandemic, and then when the pandemic occurred, we repeated the data collection with identical measures for both conditions. Also, we conducted qualitative analyses of reflections on how instructors have adapted, developed, and conscientized their teaching to deal with and positively help students in the times of crisis during the pandemic.

**Hypotheses.** We tested three different hypotheses in this study.

1.  There is a statistically significant difference in the nature and type of positive experiences between the Before COVID-19 and During COVID-19 groups.
2.  There is a statistically significant difference in the frequencies of the types of feelings and emotions between the Before COVID-19 and During COVID-19 groups.
3.  There is a statistically significant difference in the level of positive perceptions felt between the Before COVID-19 and During COVID-19 groups.

## 2. Materials and Methods

### 2.1. Sample

The sample comprised undergraduate students in a state university in Florida. The before "course crafting" group (Before COVID-19) had 155 respondents, but there were only 137 usable responses. There were 51 males, 82 females, and 4 people who identified as another gender identity. For the same group, 69% of the sample was between the ages of 18 and 25 years, 26% were between 26 and 30, 4% were between 30 and 45, and 1% were over 45. The students in the course crafting group (During COVID-19) were not the same students as those in the Before COVID-19 group. The During COVID-19 group were taking the same course taught to the Before COVID-19 group in the subsequent semester. The During COVID-19 group was taught online. There were 151 respondents, but only 140 responses were usable. There were 48 males, 89 females, and 3 people who identified as another gender identity. Of this sample, 64% were between the ages of 18 and 25, 29% were between 26 and 30, 6% were between 30 and 45, and 1% were over 45.

At the time of the study, we were exploring the positive experiences of students, and then, once the COVID-19 pandemic occurred, we continued this study as a quasi-experimental study. Although this study was designed as a cross-sectional study, the situation that arose allowed for a semi-quasi-experimental design. We called it semi-quasi-experimental, as we had no control of extraneous factors as the epidemic brought unexpected threats and opportunities that were necessary for delving into and comparing student experiences before the pandemic and during the pandemic. Also, the students in the Before COVID-19 group were not the same students who were in the During COVID-19 group; the strict and pre and post design found in traditional quasi-experimental designs was not the case. However, the experiences of both groups are comparable as the study was conducted over the same year, where the Before COVID-19 group was administered the questionnaire one semester before the pandemic, and then the During COVID-19 group

was administered the questionnaire during the pandemic. They were all from the College of Arts and Sciences and faced the same pandemic environment.

### 2.2. Instruments and Procedure

A critical incident open-ended method that has been used by authors in other studies was modified and adapted for this study [24–26]. The open-ended qualitative data collected with this inductive approach captured the unique conditions and experiences that the students themselves perceived as positive. However, there could be some limitations in the data obtained because of memory distortions and personal perspectives. For example, the time between the Before and During COVID-19 groups was a very short period, i.e., one semester. Students were also asked to be specific and concrete and had the opportunity to say none if they experienced no positive experiences. We also looked at the inter-rater reliability to look for consistencies in the themes that emerged in order to obtain reliable data from this method.

We obtained Institutional Research Board (IRB #1548579-1) approval to perform this study at a state university in Florida. All surveys were administered by the Qualtrics survey method and were completely online, facilitated by the Institutional Research office in the university. The surveys were voluntary and completely anonymous. The critical incident method used by researchers in several studies in management asked students to briefly describe any event, episode, or incident related to their academic or specific class experience that made them feel challenged, energized, motivated, or excited, and what made them feel positive in some way. As they described this experience, they were also asked to explain why and how this experience made them feel positive. Finally, they were also asked to rate how positive they felt about the experience on a 4-point scale. If they had no such experience, they could say "none" and exit the survey. Consent to participate was obtained by using an electronic consent form in our Qualtrics survey.

In the second group (During COVID-19), the students were given the same instructions but asked to rate their experiences for the classes they were taking that had transitioned to a remote format. All of these students were from the College of Arts and Sciences, taking behavioral sciences courses, and taking classes in the spring semester of 2020.

### 2.3. Data Analysis

Content analysis was performed on the responses given by the Before COVID-19 group and During COVID-19 group. Graduate students were trained for two weeks in the content analysis process before examining the qualitative data collected from students: (1) describing their most positive experience and (2) how and why they felt so positive. These responses were classified by two raters based on the themes that emerged, and those with similar themes combined took the thematic-lead classification. Once these frequencies were established, a non-parametric chi-squared test was run to determine the homogeneity statistics and whether significance was shown between the two groups, Before COVID-19 and During COVID-19.

The analysis was both qualitative, using content analysis, and quantitative, as we used the chi-squared test to look at the frequencies of the types of positive experiences and the feelings and reactions to these experiences. We examined data on positive experiences three months before COVID-19 and then we compared them to the positive experiences during COVID-19. The intention was to link the course crafting critical incidents with the ways they imputed positive experiences among students.

The categories/themes that emerged were looked at by having each rater independently look for ways to describe the incidents that emerged. The categories were sorted to identify similar themes, as is often the method in content analysis. Categories that were closely related with common themes were combined into higher-order categories. Any disagreements were resolved by consensus. A measure of the inter-rater reliability was calculated by having a third rater code them again to look for consistency and to obtain a measure of reliability. The third rater, who was blind to the hypotheses and was not

involved in the original development of the categories, attempted to match the incidents to the original categories.

## 3. Findings

### *3.1. Inter-Rater Agreement*

The inter-rater agreement of the first two raters with the third raters came to 92–98%. Some classification disagreements were handled appropriately through a rater consensus. Disagreements between the two main raters were handled by a third rater, and the percentage of times a third rater classified these incidents correctly was high. Only 137 of the 155 responses were usable, because 11% of the first group (Before COVID-19) indicated "none" and stated that they had not had a particularly positive experience. In the second group (During COVID-19), 7% of respondents indicated "none", with 140 valid responses. The first result is reported in Table 1. The positive experiences are listed in the first column alongside the frequencies of their occurrence for the Before and During COVID-19 groups. The topics and frequencies of the two groups are shown in Table 1. The main purpose of this analysis is to understand what the positive experiences were and why the groups had such positive experiences.

**Table 1.** Main themes and frequencies of positive experience responses before and after course crafting, with a sample response for each category.

| Category Types of Positive Experiences | Before COVID-19 | During COVID-19 |
|---|---|---|
| Real-life connections "Our assignments had a great connection to real-life experiences and the practical approach was very useful, I appreciated the applied examples" | 20 | 7 |
| Exciting research project/Assignment "The project was very interesting and the high standards and expectations made me work very hard and I am developing so many skills" | 12 | 9 |
| Faculty characteristics "The knowledge and expertise of the professor were inspirational and made me work more and want to learn. She is terrific" | 23 | 7 |
| Personal and emotional support "I was so touched by the personal care, empathy, kindness, and emotional support I received from many of my instructors" | 11 | 23 |
| Instructors' constant, prompt feedback and availability "I was reassured and gratified by the cheerful presence and reassurance of my instructor, who encouraged me to reach out whenever I needed her" | 9 | 17 |
| Selflessness, dedication, and commitment "I now have an increased value and appreciation for [what] my teachers do and this semester I will always have the fondest memory of this university as these experiences have been most positive" | 9 | 21 |
| Creativity and challenge "The experience of being challenged in the class and being empowered was very motivating" | 20 | 8 |
| Self-reflection and realization "The class experiences made me self-reflect and I learned not just the subject, but more about the value of life and happiness and joy in little things" | 7 | 17 |
| Sense of accomplishment and confidence "Freedom and independence in the assignment of independently creating a website gave me a great sense of accomplishment" | 17 | 8 |

**Table 1.** *Cont.*

| Category<br>Types of<br>Positive Experiences | Before COVID-19 | During COVID-19 |
|---|---|---|
| New perspective and approach<br>"Taking this class now and the experiences and interactions have been enlightening, I learned to live [in] the moment, slow down, chill, appreciate more and complain less, and never take things for granted" | 9 | 22 |
| Total | 137 | 140 |

*3.2. Positive Experience*

Hypothesis one, which predicted that there would be a statistically significant difference in the nature and type of positive experiences between the Before COVID-19 group and the During COVID-19 group, was supported. We used the SPSS software package and examined the frequencies of responses using a chi-squared test of homogeneity to determine whether the proportions of the types of positive experiences were different between the two groups (Before COVID-19 and During COVID-19 groups). The proportions of the different types of positive experiences were significantly different between the two groups: $\chi^2(9, 276) = 44.71$, $p = 0.001$.

The most important difference between the two groups was the nature of the experiences during the two periods. Before COVID-19, students reported frequent experiences related to faculty's expertise characteristics (n = 23), real-life connections (n = 20), creativity and challenge (n = 20), and having a sense of accomplishment and confidence (n = 17). In the During COVID-19 group, the most frequent positive experiences reported by the students included personal and emotional support (n = 23), new perspective and approach (n = 22), and selflessness, dedication, and commitment (n = 21).

*3.3. Emotional Experience*

A second analysis was performed to address whether there were differences in the feelings and emotional reactions between the Before and During COVID-19 groups. Hypothesis 2 was supported with statistically significant differences in the frequencies of the types of feelings and emotions reported across the two groups (Before COVID-19 and During COVID-19). There were also significant differences in the reactions reported between the two groups: $\chi^2(8, 277) = 42.62$, $p < 0.001$. Table 2 presents the different feelings and emotional reactions to these experiences.

**Table 2.** Main feelings and reactions and their frequencies, with a sample response for each category.

| Category<br>Types of Feelings and Emotional Reactions | Before COVID-19<br>(Time 1) | During COVID-19<br>(Time 2) |
|---|---|---|
| **Self-confident**<br>"I feel so good about myself. Confidence in my ability and knowledge is very high" | 21 | 9 |
| **Proud and Joyful**<br>"I am proud and thrilled about my performance in the class" | 19 | 8 |
| Motivated and Energetic<br>"I looked forward to this class every day and I am motivated and excited about the class" | 22 | 8 |
| Resilient and strong<br>"There is a thrill of success amidst chaos and in the face of insurmountable odds" | 12 | 25 |

| Category<br>Types of Feelings and Emotional Reactions | Before COVID-19<br>(Time 1) | During COVID-19<br>(Time 2) |
|---|---|---|
| Uplifted, blessed, and thankful<br>"I am appreciative of the kindness and care that I have experienced, and this really touches my heart and moves me" | 8 | 23 |
| Grateful and connected<br>"I have never felt such a great connection as I have to all my teachers and I realize how selfless they are, how flexible and understanding, and [how] caring they are" | 9 | 24 |
| Hopeful and inspired<br>"I am very hopeful that I will do well in this career path and I am inspired to learn" | 21 | 10 |
| Valued and appreciated<br>"I feel so valued and appreciated, I have been given so much attention. All my communications have been so emotional but my instructor has been open and listened to me" | 7 | 17 |
| Resourceful and independent<br>"I have developed so many resources and learned so many new skills and I am so happy I took this class" | 18 | 16 |
| Total | 137 | 140 |

The Before COVID-19 students reported a higher number of feelings; they were motivated and energetic (n = 22), self-confident (n = 21), and hopeful and inspired (n = 21). On the other hand, for the During COVID-19 group, the most frequent reactions reported by students included feeling resilient and strong (n = 25), grateful and connected (n = 24), and uplifted, blessed, and thankful (n = 23).

*3.4. Intensity of Positive Experiences*

Hypothesis 3 predicted that there would be a difference in the mean scores of the intensity of positive emotions felt between the two groups. This hypothesis was also supported by the reported data. We looked at the intensity level of the emotions of the students, who were asked to rate how intensely they felt each reported emotion on a scale of 1–4 (1 being the 'least intense' and 4 being the 'most intense'). There was a significant difference in the mean intensity of emotions felt between the Before COVID-19 (M = 1.56, SD = 0.76) and During COVID-19 (M = 2.86, SD= 0.76) groups. A t-test was run and found significant differences in the intensity of feelings and emotions, with a greater intensity of feelings in the During COVID-19 sample (t = 3.06, df= 375, $p < 0.0001$) at a statistically significant level.

There were some specific examples and descriptions of changes made in the course after the "course crafting" process that students felt were very positive experiences, and details of these are presented in (Table 3).

As discussed in the introduction, this study attempts to understand how educators were "crafting" their courses to make them a good fit for the transition to distance learning and by changing or modifying content. We used the three theoretical categories of the job crafting model to understand how these courses were modified at the task level, the relationship level, and the thought level.

In Table 3, we give examples of how this was carried out. We observed that many activities were changed to make them individualized to align with student needs. For example, there were gratitude exercises for students to help them think about life and relate to the course content by using the crafting process. So, students were required to be reflective in essay questions and made connections to the pandemic crisis. Another example of crafting was how instructors or teaching assistants connected to students, such as by

reaching out to students personally via Zoom, emails, or phone calls when needed. There were also Zoom assignments that encouraged students to share their personal experiences of the crisis and relate this to many of the course's objectives.

**Table 3.** Examples and descriptions of changes made in the courses at the task, relationship, and thought levels.

| | |
|---|---|
| Examples of changes in the nature and types of the tasks, assignments, and homework | 1. Many self-reflection assignments, journaling, and assignments that relate to the present environment such as gratitude exercises and exercises on coping with stress.<br>2. Options to change assignments to fit the context, more latitude and autonomy, and individualized projects were given to students to "craft" some examples of topics they wanted to write on that would be relevant or meaningful for them and that were tied to course objectives.<br>3. Many exams were conceptual essays, were open-book, were reflective and connected to some of the experiences related to the crisis, and had direct meaning and relevance to current experiences.<br>4. Many exercises on self-growth and self-development and reflective papers.<br>5. Optional assignments to share inspirational stories of courage on the discussion board or Zoom meetings. |
| Examples of interactions and communication with students | 1. Cheerful and constant presence of instructors making many announcements during the semester.<br>2. Reaching out to students at a personal level frequently.<br>3. Communications via not just Zoom calls, but with phone calls to students from the instructor or teaching assistant.<br>4. Being available to students' evenings and weekends and linking students to different support services throughout the community.<br>5. Checking on students who were absent for a few classes consistently with direct interaction.<br>6. Activities and group work to increase interactions with other students in periods of social isolation.<br>7. Greater empathy and understanding, willingness to accept late assignments, and being flexible with deadlines. |
| Examples of how students perceived these changes and their thoughts and feelings | 1. Students felt more valued and respected and expressed a new appreciation for university life in general; classes were not just about grades, they were about seeing the bigger picture.<br>2. Gaining a new view of life, living in the moment, slowing down, chilling, appreciating more and complaining less, and never taking things for granted. |

### 3.5. Course Crafting Examples

The new task crafting examples included different tasks such as self-reflection assignments, journaling, and assignments that relate to the current environment, including exercises on gratitude or exercises on coping with stress. There were also tasks that were optional assignments: these included inspirational stories of courage presented on the discussion board or in Zoom meetings. Students also discussed relational crafting in terms of the positive changes that the online communication medium offered. These included the cheerful and constant presence of instructors making many announcements during the semester and the asynchronous meetings. The instructors were also appreciative, providing almost immediate emotional support, empathy, kindness, reassurance, and encouragement when needed.

Finally, in terms of thought crafting, examples of students' perception of the class, the instructor, and the university show that the students expressed a new appreciation and perspective of the university and life in general. Students emphasized the tone of class climate and how this was very inspirational and motivating; there was a sense of being close together, as some students claimed that "we are all in this together".

## 4. Discussion

This study focused on the experiences that students perceived as positive before and during the COVID-19 pandemic. The findings provide some valuable insight into what students perceive as positive and affective experiences in a time of emotional perturbations. In this study, we were specifically interested in how students responded to the transitioned courses and the effectiveness of the course crafting process. We specially examined what were some of the changes made and what the students found positive.

Changes at the task level, the relational level, and the thought level (cognitive) were all related to the positive experiences students had. It seems like even a small gesture or a reassuring behavior can have a very powerful impact on a student when faced with isolation and tempestuousness. Another very important finding is the role of emotions. Whether it is implicit or explicit, emotions are heightened when loneliness and uncertainty are pervasive, and compassion or gratitude can have a considerable impact on the wellbeing of individuals. Student emotions were moderated by relational crafting through perpetual extended lines of communication with the instructor. The importance of emotional support was very evident in the During COVID-19 group. The pandemic had opened a new emotional realm for students that did not exist pre pandemic. The sort of caring seen in the post-pandemic empathy, kindness, selflessness, and dedication to students was not felt or seen before in the classroom.

There were significant differences in the themes that emerged: first, in the nature of the students' perception of their positive experiences and second, in the emotions and feelings they experienced. There was also a significant difference in the intensity of the emotions felt between the two groups. These intensities appear to have emotional perturbations among the During COVID-19 group, showing higher levels than those who were in the face-to-face classroom (the Before COVID-19 group). Academics have shown agility, concern, and accommodation for students and were able to reach out to students and provide a positive psychological intervention of support and empathy.

### 4.1. Course Crafting

The results give insight into how the crafted courses and communications media engaged and made the course more meaningful. Although we cannot claim that the course crafting process changed the perceptions of students in terms of their positive experiences as this was not an experimental study, it is interesting to examine the descriptions of students' perceptions of the process.

The COVID-19 pandemic presented a unique opportunity for understanding the course crafting process when altering courses using the theoretical underpinnings of the job crafting model in an educational context. Specifically, university instructors, content developers, course designers, technical experts, and student advisors worked and put together courses to transition into a remote learning format. Another way to interpret these positive experiences was to use the theoretical framework of the job crafting model. This concept has been applied in many areas of positive organizational psychology, which looks at how to improve the workplace environment. In the examples presented in Table 3, it appears that there were many changes at the task, relationship, and cognitive levels as "course crafting" took place when the courses were transitioned during the pandemic. These changes in the redesigned course pedagogies worked as this job crafting was carried out in these times of crisis to make the courses more meaningful, motivating, and engaging for students in this new virtual format [20,21]. Furthermore, the job crafting model has been used to understand other areas, including career crafting and team crafting [19,27].

### 4.2. Pre and Post COVID-19

Given the uncertainty of the COVID-19 pandemic and the many stressors and challenges individuals faced, whether psychological, mental, physical, financial, or behavioral, educators had to do many things. They had to shuffle between recreating the pedagogies of their courses and changing their mediums of instruction, and providing the support

for students' needs in the pandemic environment. Other research studies also emphasize the importance of maintaining awareness of students' socio-emotional needs in the online environment [28].

The emotional issues and reactions of the During COVID-19 group showed the intensity of the factors that arose during COVID-19 circumstances. It was reassuring to observe so many favorable viewpoints in the second group, and many of the comments went beyond academic or class considerations. In contrast to the Before COVID-19 group, the During COVID-19 group experiencing the pandemic had more frequently positive perceptions of academic, social, and emotional support. Some of these results should be interpreted with some caution, as the groups are different; the Before COVID-19 students were not the same as those in the During COVID-19 group.

Several aspects of this study are directly related to the changes faculty members made through crafting their courses. The faculty members, rather than researching tools and technologies to deal with the new "social distancing" reality, began to acclimate to the problems students face, at the same time emphasizing new pedagogies fitting student needs [29]. Further, this study underlines how crucial it was to comprehend the affective element in this "new normal" environment, as well as the significance of being more adaptable to the requirements of learners in the future, to show care and trust, and to be open and supporting [30]. Thus, a sole concentration on technology won't solve the affective issues, whether it is isolation, loneliness, anxiety, or even depression.

Course crafting demonstrates numerous factors at play, as is evident in this study. One obvious change was the agility of the faculty, maintaining professional standards at the same time as becoming flexible, demonstrating humaneness and thoughtfulness, and being caring. The relational crafting showed that the faculty had to instill new professional behaviors such as sharing, empathy, increased listening dispositions, sensitivity, understanding students' personal situations, and showing genuine concern of what people are going through. The beliefs and assumptions about educators' limitations, including the stereotype of cold professionalism and rigid boundaries, may not be appropriate in this type of crisis environment.

### 4.3. Limitations

This study has many limitations, as many of the student experiences may have been conditioned and mediated by many other variables unique to the pandemic climate. We understand that students faced isolation, loneliness, life monotony, and despair, and were only interested in their affective academic experience. For understanding students' perceptions during COVID-19 and instructional changes, rather than delving into their negative perceptions, the goal of this study focused on positive reactions. This was a very small sample, and there may have been many differences in the two groups. Although we gained some valuable knowledge of what students find positive in the online environment in a time of crisis, we must be cautious of generalizing these findings based on this small study's data.

This study was a microcosm of what we are seeing and may be representative of what academia has been experiencing in the aftermath of the pandemic and around the world. We have all experienced this profound nearness to students during the pandemic, not only in the United States, and not only among students but also in other parts of the world that faced the pandemic and especially among heath care workers [31]. This study may provide many insights into what happens in situations of crises in a student population.

### 4.4. Future Directions, Implications, and Conclusions

This study investigates the positive perceptions of students during a time of crisis and how students choose to focus on positive experiences. Our study sheds light on the most effective and necessary digital-affective strategies to teach and learn. This gave the students a better understanding of the specific conditions and situations that they identified as positive in teaching and learning environments. Many of the tasks, relationships, and

thought process strategies appear to have increased the course's relevance, motivated students, and made them more satisfied in the face of chaos. Given the range of emotional experiences, including stress, that students are experiencing because of the pandemic, there is certainly a need for emotional support and a need to focus on supporting students academically. It is underlined that positive psychology encourages compassionate, kind, flexible, encouraging, and hopeful experiences. Students within these circumstances and with this support are becoming tenacious and eager to improve.

This study contributes to the existing knowledge in helping to understand the unique conditions that we face in a crisis, and the conditions that lead to positive experiences during turmoil. This will assist in better understanding the social impact of COVID-19-like conditions and how to better facilitate learning. This study added to our knowledge of the benefits of focusing and understanding positive experiences in a crisis. In the face of adversary or difficult times, people can maintain a positive attitude, complete gratitude exercises, develop self-discovery exercises, engage in self-reflection and count their blessings, and other novel tasks to overcome numerous challenges [32].

**Author Contributions:** L.N. conceived the idea for such a study, came up with the rationale, wrote the bulk of the paper, and came up with the research questions. R.N. wrote, edited, and revised the manuscript thoroughly, and looked over the data analysis and data reporting. S.M. contributed to the literature review, helped to identify relevant past research, edited the paper many times, and came up with ideas on how to present the data. B.W. conducted the data analysis and helped with formatting the paper, including the references. All authors have read and agreed to the published version of the manuscript.

**Funding:** Open Access funding provided by the Qatar National Library.

**Institutional Review Board Statement:** This paper has been through the IRB process and was judged as exempt, with an IRB number of IRB# 1548579-1 from the University of North Florida.

**Informed Consent Statement:** Written informed consent has been obtained from the patients to publish this paper.

**Data Availability Statement:** The data presented in this study are available on request from the corresponding author. The data are not publicly available due to privacy.

**Conflicts of Interest:** The authors declare no conflict of interest.

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
