# Peer review of "Course Crafting in the Pandemic: Examination of Students’ Positive Experiences"

_education, doi:10.3390/educsci14020131_

Round 1
Reviewer 1 Report
Comments and Suggestions for Authors
The results of the research described in this paper are interesting for the educational community and the knowledge is relevant for the improvement of teaching and learning in higher education in general. The paper is well written, the structure is good, and the literature cited is relevant.
Please check the sentence in the row 79-80:
It is suggested he maintenance of positive attitude in the face of trying times
and experiences can help students become more resilient.
Please check the text in the rows 333, 334, and 335. This text is not connected to the paper:
his section may be divided by sub- (333) headings. It should provide a concise and precise description of the experimental results, (334) their interpretation, as well as the experimental conclusions that can be drawn. (335)
Author Response
Dear Sir,
Thank you for reviewing the manuscript your insight have made us look at the manuscript and ameliorated accordingly. We include the reviewer comments and responses shaded in yellow.
The results of the research described in this paper are interesting for the educational community and the knowledge is relevant for the improvement of teaching and learning in higher education in general. The paper is well written, the structure is good, and the literature cited is relevant.
Please check the sentence in the row 79-80:
It is suggested he maintenance of positive attitude in the face of trying times
and experiences can help students become more resilient.
We have edited the sentence.
Please check the text in the rows 333, 334, and 335. This text is not connected to the paper:
his section may be divided by sub- (333) headings. It should provide a concise and precise description of the experimental results, (334) their interpretation, as well as the experimental conclusions that can be drawn. (335)
We changed to the Results section to the Findings section. We also created subsections to the Findings Sections:
3.1 Interrater Agreement
3.2 Positive Experience
3.3 Emotional Experience
3.4 Intensity of Positive Experiences
See paper attached
Reviewer 2 Report
Comments and Suggestions for Authors
The method and conclusions should be more precisely justified by the authors.
Author Response
Dear Sir,
Thank you for reviewing the manuscript your insight have made us look at the manuscript and ameliorated accordingly. We include the reviewer comments and responses shaded in yellow.
The method and conclusions should be more precisely justified by the authors.
We have reviewed and edited the manuscript and the necessary changes. The method and results section have been classified into sections and edited extensively. We have attached the manuscript for your review.
Reviewer 3 Report
Comments and Suggestions for Authors
This paper aims to analyze students’ perceptions regarding a course crafting during COVID-19 and attempts to make comparisons with the perceptions of another group of students before covid at the same university.
The idea is quite interesting but authors should examine the following issues:
Background:
1. Line 60: I am wondering whether “strengths and meaning” are taxonomized as positive feelings. Maybe the authors tend to refer to satisfaction? Clarification is needed there.
2. Line 68: The reference does not follow the numerical system.
3. Lines 74-76: Syntactical issue. The subordinate sentence (When…epidemic) is not well structured without a main sentence.
4. Subsection 1.2. Positive Psychology, Job Crafting and the Pandemic: Authors describe how job crafting is adapted to course crafting. I think this part should be restructured since the terms appear without clarification and are mingled in a way. I mean that they start from job crafting, describing then jump to course crafting and returning back to job crafting. I believe this is confusing for readers.
5. Lines 121-122: Syntax issues. Please restructure.
6. Lines 130-131: Sentences should separate.
7. Lines 135-136: The sentence is incomprehensible. Please restructure.
8. Authors refer to critical incidents technique and describe it briefly. I think that a thorough description of the method’s core elements should be included and they should also refer to its limitations (taking into consideration that memory issues or personal lenses always interfere here). It is important since they employed this technique for data collection.
Materials and Methods
1. Participants should be mentioned in a different subsection. I have serious consideration since authors are referring to comparisons BUT the groups consist of different students. In this light, how can we take into account the experiences of the Before COVID-19 group and compare it with the other group? Authors should necessarily explain why this works. Additionally, besides gender, I think that the semester the students attended should be included. Their age may suggest something as well.
2. Authors do not actually describe the instruments they used. Was it an open-ended questionnaire? Did they use reflection diaries? Was it field notes? Why do they include information about the analysis in the subsection: Instruments? I think these should be included in a different: Data analysis. Additionally, authors refer to content analysis, but they do not refer to the inductive approach that I believe they had followed. In the method section
3. Authors should include information regarding participants’ consent for their participation in the research.
4. Subsection: 2.3 Procedure: Information about the critical incident method should be presented in the instrument subsection. Additionally, authors should be consistent regarding the terms: is it a method or a technique as it appeared in the background?
5. Authors refer to inter-rater reliability but do not give information regarding Cohen’s kappa. This information is offered in the results section. I think this should be included earlier in the methods section.
The whole section should be restructured with better and more precise organization between sections. I believe that authors should focus on clarity in order to present their research in a more organized way.
Results
1. The categories that arose from content analysis should be described. I mean which answers were coded there is something that we need to know. Further, quotes given from the participants should be accompanied by participants’ gender or an undercover name for a more realistic representation. This should be applied in every table this quote appears.
2. I believe that in table 3, there are too many quotes presented while the categories are not described. Further, I believe that more information regarding the way the course was crafted should included. I mean the ways the professors decided to reorganize it during the pandemic. I, also, understand that more categories arose from the question of ‘why and how they felt positive’. This should be explicitly mentioned.
3. Information regarding the course crafting (to which I referred earlier) appears out of the blue in the results section while readers should have more information about it earlier, in a separate subsection.
Discussion
1. Discussion part should be supported by contemporary literature.
2. Authors claim that: “The importance of emotional support is very evident of the During COVID-19 group. The pandemic had opened a new emotional realm for students that did not exist pre-pandemic.” However, since the participants of the two groups are different how can we compare their perceptions in that field? I mean, what if Before COVID-19 group consists of more resilient participants?
3. Lines 333-335: Authors have left the instructions of the journal in their text. Please, pay particular attention to the whole text since it needs more care and careful reading.
4. Lines 337-339: Please restructure.
5. Lines 370-371: I do not believe that authors can claim that course crafting brought the changes in the results since participants are different. Maybe they can solely emphasize the elements of the course without further comparisons.
6. Line 391: what about the dollar sign?
7. Lines 392-393: Please restructure.
8. About the limitations: the authors should refer to and explain the different groups. This is very important.
9. I think that I was surprised by the introduction of the terms resilience and resilient at the end of the discussion part. I mean did they students become more resilient? Can the authors support that from the study findings. I think that the majority of findings suggest their satisfaction from the course as well as positive feelings they experienced.
10. Line 430: The citation does not follow the rules of the journal.
11. Are references written according to the journal’s system?
Authors should be very careful with syntax all over their text. Some sentences appear to be incomplete or subordinate sentences are not based on a main sentence. Some parts should be reorganized for clarity.
Comments on the Quality of English LanguageAuthors should carefully reread their text for syntax and grammatical issues.
Author Response
Dear Sirs,
Thank you for reviewing the manuscript. Responses are shaded in yellow under the reviews.
Background:
- Line 60: I am wondering whether “strengths and meaning” are taxonomized as positive feelings. Maybe the authors tend to refer to satisfaction? Clarification is needed there.
That was a very good point and we have changed some of the terms.
- Line 68: The reference does not follow the numerical system.
Done
- Lines 74-76: Syntactical issue. The subordinate sentence (When…epidemic) is not well structured without a main sentence.
Done
- Subsection 1.2. Positive Psychology, Job Crafting and the Pandemic: Authors describe how job crafting is adapted to course crafting. I think this part should be restructured since the terms appear without clarification and are mingled in a way. I mean that they start from job crafting, describing then jump to course crafting and returning back to job crafting. I believe this is confusing for readers.
This has been changed it is clearer now, we have also used course crafting in the subheading and explained this better.
- Lines 121-122: Syntax issues. Please restructure.
done
- Lines 130-131: Sentences should separate.
done
- Lines 135-136: The sentence is incomprehensible. Please restructure.
done
- Authors refer to critical incidents technique and describe it briefly. I think that a thorough description of the method’s core elements should be included and they should also refer to its limitations (taking into consideration that memory issues or personal lenses always interfere here). It is important since they employed this technique for data collection.
done
Materials and Methods
- Participants should be mentioned in a different subsection. I have serious consideration since authors are referring to comparisons BUT the groups consist of different students. In this light, how can we take into account the experiences of the Before COVID-19 group and compare it with the other group? Authors should necessarily explain why this works. Additionally, besides gender, I think that the semester the students attended should be included. Their age may suggest something as well.
This is now spotted all over the manuscript that it was a different group of subjects, the semester was Spring 2020 and this is included. We did not include gender as this study did not focus on gender differences and we have planned a study in the future, that will focus on this topic and we will do a comprehensive literature review on this to support gender differences if any with such positive experiences.
- Authors do not actually describe the instruments they used. Was it an open-ended questionnaire? Did they use reflection diaries? Was it field notes? Why do they include information about the analysis in the subsection: Instruments? I think these should be included in a different: Data analysis. Additionally, authors refer to content analysis, but they do not refer to the inductive approach that I believe they had followed. In the method section
We have mentioned that this is a open-ended instrument and we have addressed many of concerns mentioned here including mentioning that this is an inductive method. Thank you for pointing out these issues.
- Authors should include information regarding participants’ consent for their participation in the research.
Done
- Subsection: 2.3 Procedure: Information about the critical incident method should be presented in the instrument subsection. Additionally, authors should be consistent regarding the terms: is it a method or a technique as it appeared in the background?
Done, made is all “method” and removed “technique”
- Authors refer to inter-rater reliability but do not give information regarding Cohen’s kappa. This information is offered in the results section. I think this should be included earlier in the methods section.
We used the percentage of agreement method that we have been adopted in many previous studies that have been published in the past.
The whole section should be restructured with better and more precise organization between sections. I believe that authors should focus on clarity in order to present their research in a more organized way.
Done
Results
- The categories that arose from content analysis should be described. I mean which answers were coded there is something that we need to know. Further, quotes given from the participants should be accompanied by participants’ gender or an undercover name for a more realistic representation. This should be applied in every table this quote appears.
This has been modified.
- I believe that in table 3, there are too many quotes presented while the categories are not described. Further, I believe that more information regarding the way the course was crafted should included. I mean the ways the professors decided to reorganize it during the pandemic. I, also, understand that more categories arose from the question of ‘why and how they felt positive’. This should be explicitly mentioned.
Done.
- Information regarding the course crafting (to which I referred earlier) appears out of the blue in the results section while readers should have more information about it earlier, in a separate subsection.
We have now mentioned this earlier.
Discussion
- Discussion part should be supported by contemporary literature.
We have cited many studies to support what we found.
- Authors claim that: “The importance of emotional support is very evident of the During COVID-19 group. The pandemic had opened a new emotional realm for students that did not exist pre-pandemic.” However, since the participants of the two groups are different how can we compare their perceptions in that field? I mean, what if Before COVID-19 group consists of more resilient participants?
This is a valid point, we have changed the way we present this information.
- Lines 333-335: Authors have left the instructions of the journal in their text. Please, pay particular attention to the whole text since it needs more care and careful reading.
Removed
- Lines 337-339: Please restructure. Done
- Lines 370-371: I do not believe that authors can claim that course crafting brought the changes in the results since participants are different. Maybe they can solely emphasize the elements of the course without further comparisons.
Has been reworded.
- Line 391: what about the dollar sign?
Done
- Lines 392-393: Please restructure. Done
- About the limitations: the authors should refer to and explain the different groups. This is very important.
Done
- I think that I was surprised by the introduction of the terms resilience and resilient at the end of the discussion part. I mean did they students become more resilient? Can the authors support that from the study findings. I think that the majority of findings suggest their satisfaction from the course as well as positive feelings they experienced.
Resilience removed and sentences modified.
- Line 430: The citation does not follow the rules of the journal. Done
- Are references written according to the journal’s system?
Awkward citations were removed and corrected.
Authors should be very careful with syntax all over their text. Some sentences appear to be incomplete or subordinate sentences are not based on a main sentence. Some parts should be reorganized for clarity.
Comments on the Quality of English Language
Manuscript edited heavily.
Authors should carefully reread their text for syntax and grammatical issues.
Round 2
Reviewer 3 Report
Comments and Suggestions for Authors
Authors answered my comments. I would like to add some more mainly for clarification issues.
Lines 56-57: Syntactical and grammatical issues. Please restructure.
Line 68: I think an ‘is’ is missing there. The sentence is not comprehensible.
Line 64: Subsection: 1.1 Positive psychology and the pandemic. I think the title is generic since this subsection focuses on the students’ psychology and this is not suggested by the heading.
Lines 107-112: Authors refer to job crafting but how is this information connected with students and course crafting? I mean do these data apply for students as well? Then this should be clarified. However, I still believe that this whole subsection needs a more organized structure.
Line 144-146: Syntax issues. Please restructure.
Line 153: I do not really understand the meaning of the first sentence there. Do authors mean the critical incident method?
Lines 157-159: The sentence is too long and needs restructuring.
Line 337: is ‘algin’ a misspelling for align?
Lines 469-470: I am not pretty sure that authors can claim the following: ‘ these findings may be general-469 izable to other populations and settings, such as virtual workplaces in crisis situations’
Comments on the Quality of English LanguageSome improvement can be applied.
Author Response
Thank you so much for your comments again, we appreciate your time and effort to look at this so meticulously, it is a better document because of your feedback. Here are the changes we made in response to your comments.
Lines 56-57: Syntactical and grammatical issues. Please restructure.
DONE. This was restructured.
Line 68: I think an 'is' is missing there. The sentence is not comprehensible.
DONE. This was restructured, the “is” added and sentence corrected.
Line 64: Subsection: 1.1 Positive psychology and the pandemic. I think the title is generic since this subsection focuses on the students' psychology and this is not suggested by the heading.
The title has been changed, thank you for that excellent suggestion.
Lines 107-112: Authors refer to job crafting but how is this information connected with students and course crafting? I mean do these data apply for students as well? Then this should be clarified. However, I still believe that this whole subsection needs a more organized structure.
We have now given more clarification.
Line 144-146: Syntax issues. Please restructure.
DONE
Line 153: I do not really understand the meaning of the first sentence there. Do authors mean the critical incident method?
This has been clarified.
Lines 157-159: The sentence is too long and needs restructuring.
DONE
Line 337: is 'algin' a misspelling for align?
Yes we corrected this.
Lines 469-470: I am not pretty sure that authors can claim the following: ' these findings may be general-469 izable to other populations and settings, such as virtual workplaces in crisis situations'
This has been rephrased, that was an excellent suggestion.
A clean copy is attached.